# Beyond Screen Time: Exploring the Associations between Types of Smartphone Use Content and Adolescents’ Social Relationships

**DOI:** 10.3390/ijerph19158940

**Published:** 2022-07-22

**Authors:** Shunsen Huang, Xiaoxiong Lai, Xinmei Zhao, Xinran Dai, Yuanwei Yao, Cai Zhang, Yun Wang

**Affiliations:** 1State Key Laboratory of Cognitive Neuroscience and Learning, Beijing Normal University, Beijing 100875, China; huangss@mail.bnu.edu.cn (S.H.); xxlai@mail.bnu.edu.cn (X.L.); zhaoxm@mail.bnu.edu.cn (X.Z.); daixinran@mail.bnu.edu.cn (X.D.); 2Department of Education and Psychology, Freie Universität Berlin, 14195 Berlin, Germany; yywyao@gmail.com; 3Einstein Center for Neurosciences Berlin, Charité—Universitätsmedizin Berlin, 10117 Berlin, Germany; 4Berlin School of Mind and Brain, Humboldt-Universität zu Berlin, 10117 Berlin, Germany; 5Collaborative Innovation Centre of Assessment toward Basic Education Quality, Beijing Normal University, Beijing 100875, China; zhangcai@bnu.edu.cn

**Keywords:** types of smartphone use content, social relationships, adolescents, specification curve analysis

## Abstract

The past two decades have witnessed controversy over whether the use of digital technology has damaged or enhanced adolescents’ social relationships, which influences their development. In this study, we addressed this debate by specifying the effect of different types of smartphone use content on social relationships, rather than simply relying on screen time spent on digital media. To avoid selective analysis and report of different variables, we used specification curve analysis (SCA) in a large dataset (N = 46,018) to explore the correlations between 20 types of smartphone use content and adolescents’ social relationships (parent–child, peer, and teacher–student). The types of smartphone use content were measured by the revised version of Mobile Phone Use Pattern Scale, the Parent-Child Relationship Scale, the Peer Relationship Scale, and the Teacher-Student Relationship Scale assessed three different social relationships, respectively. Of the 20 types of smartphone use content, only playing games (negatively explaining 1% of the variation), taking online courses (positively explaining 1.6% of the variation), using search engines (positively explaining 1.2% of the variation), using a dictionary (positively explaining 1.3% of the variation), and obtaining life information (positively explaining 1.5% of the variation) showed a significant effect size. The association between smartphone use and adolescents’ social relationships depends on the various types of content with which adolescents engage during smartphone use. The various effects of different types of smartphone use content deserve the attention of both the public and policy-makers.

## 1. Introduction

Social relationships are of crucial importance in affecting individuals’ physical and mental health in the short- and long-term [1]. When children grow toward adolescence, they gradually obtain their independence and attempt to interact with broader agents in society (e.g., their peers) [2,3,4]. Parents, peers, and teachers are three significant agents among adolescents [5,6]. Maintaining good relationships with them would benefit adolescents’ social adjustment and development [7]. However, with the rapid advent of the digital age, adolescents’ social relationships are heavily influenced by the internet or other digital media technologies. Researchers, the public, and policy-makers are faced with the question of whether using digital technology helps or harms adolescents’ interpersonal interactions and relationships.

Researchers have proposed two theories to explain the effects of digital technology use on adolescents’ interpersonal relationships. Displacement theory suggests that the use of digital technologies (e.g., information communications technology [ICT], social media, smartphones) may harm individuals’ social relationships since the time and resources they previously used to maintain offline connections are replaced by online interactions [8]. For example, time spent on the internet comes at the expense of time spent on social activities, reading, and hobbies [9], which threatens their social bonds [4,10,11]. However, studies on displacement theory have shown mixed results (e.g., social media use is not related to future social interactions) [12]. Alternatively, the enhancement theory of technology use implies that technology may benefit individuals’ offline interpersonal relationships [13], which is supported by some empirical evidence [14,15]. For example, the use of digital media enables adolescents to have more opportunities to share information about themselves and spend more time with their friends, which benefits their social relationships [16]. Some qualitative literature also suggests that digital media may benefit adolescents’ social relationships, such as teacher–student [17] or parent–child relationships [18]. Thus, there is an enduring controversy regarding the link between technology use and interpersonal relationships [19,20,21].

These contradictory findings, which have led to current debates, may be due to several reasons. First, previous researchers have tried to speculate on the associations between digital technology use and interpersonal relationships mainly from the perspective of screen time; that is, the amount of time spent using technology [12,15,22,23,24,25,26,27]. However, many researchers have recently argued that when exploring the effect of the use of digital technology, researchers should not focus on screen time, as it is no longer a valid construct (rather than the time, content may be more important) [28,29]. Instead, researchers should pay more attention to the types of content of digital technology use [19,30], rather than investigating the effects of technology as a whole [31]. Empirical research found that different types of technology use contents have different influences (positive or negative) on individuals’ development (e.g., mental health) [32,33,34,35,36]. The same is true of adolescents’ interpersonal relationships. For example, some researchers have distinguished the effect between communicative (e.g., active interaction and engagement through talking, texting, and commenting) and non-communicative (e.g., passive activities such as reading, observing, and following other people’s profiles without engagement) smartphone use, and found the former to be positively associated with interpersonal communication, while the latter has been negatively tied to interpersonal communication [19]. Nevertheless, they did not elaborate on the types of content in smartphones. Recent researchers have found that communication and entertainment of technology use were associated with more negative interactions with parents, while using technology for creating content (e.g., posting on social media, creating videos) was associated with less positive interactions with parents [27]. Currently, few studies have directly explored the relationship between types of smartphone use content and social relationship and some studies just examined a few types of technology use. Second, the researcher’s arbitrary or subjective biases may influence findings, as some researchers might choose more significant results based on their subjective demands [37]. Such a phenomenon may also exist in the studies of digital technology use and interpersonal relationships. For instance, in regard to interpersonal relationships, different researchers may only report the results of one type of relationship (e.g., peer, teacher–student relationship, or parent–child relationship) [27,38,39,40]. Third, although past research has described a negative or positive correlation between digital technology use and social relationships, it remains unclear whether such effects are important enough to attract the attention of the public or policy-makers. Researchers should report on the magnitude of the effect or associations when attempting to investigate emergent technologies, as researchers have argued [31], which previous studies have rarely covered. Fourth, the social shaping theory of technology claims that the impact of technology should not merely be asserted as displacement or enhancement, but that the effects of technology use are a mixture of “affordances” and that people may use technology unexpectedly based on their choices [41]. This indicates that adolescents can choose which types of content to use when using smartphones.

Previous literature indicated that socioeconomic status (SES) may influence adolescents’ smartphone use and interpersonal relationships [42,43,44]. Screen time was found to be related to social relationships [23,27,39]. Besides, different gender has different associations with social relationships. For example, girls were found to have a better teacher–student relationship [45]. Previous studies have also shown that males and females have different orientations in smartphone use and that people of different ages also exhibit different forms of smartphone use [42,46]. Given the importance of these variables, SES, screen time spent using smartphones per day, gender, and grade are included as covariates.

Advances in smartphone technology may exacerbate the debate between displacement and enhancement theories as smartphones become increasingly multifunctional and contain a large variety of content that may have different effects on adolescents. Therefore, it is necessary to investigate the association between different types of smartphone use content and three important social relationships for adolescents (parent–child relationship, peer relationships, teacher–student relationships). On this basis, we aimed to address the debate between displacement theory and enhancement theory. Additionally, to mitigate researchers’ arbitrariness in selecting testing variables, we used specification curve analysis (SCA) to explore the associations of 20 different types of smartphone use content and social relationships. Given the above considerations, we proposed two hypotheses:

**Hypothesis** **1** **(H1).***The types of content related to interpersonal interactions during smartphone use are positively tied to adolescents’ interpersonal relationships*.

**Hypothesis** **2** **(H2).***The types of content focusing on entertainment, online transaction, taking online courses, and using utility tools are negatively tied to adolescents’ interpersonal relationships*.

## 2. Materials and Methods

### 2.1. Participants and Procedure

We drew the participants from the “2019 Regional Assessment of Educational Quality (2019 RAEQ)” dataset. The dataset was gathered in a two-stage stratified sample design in October 2019 from one Chinese eastern province, two central provinces, and one western province. Based on the basic requirement of stratified sampling for sample size [47] and the need of the local bureau of education, 85% of the school in every district was selected, which resulted in 293 schools. Students completed the questionnaires independently in the class after listening to the teachers’ instructions. In the dataset, 73,491 adolescents were surveyed; 26,681 adolescents who did not own a smartphone were excluded, and 792 participants were excluded due to incomplete questionnaires. The participants used in the final analysis included 46,018 individuals (grade 4 = 11,440, grade 8 = 34,578; male = 23,946). Table 1 contains the characteristic of the participants. The mean missing rate for each variable in the 2019 RAEQ was 4.15%. Little’s missing completely at random (MCAR) test showed that data from the 2019 RAEQ (χ^2^ = 26392.177, *df* = 2649, *p* < 0.001) were not MCAR. Missing data were processed using the EM algorithm [48]. See Figure 1 for the research methodological procedure. 

### 2.2. Measurements

#### 2.2.1. Types of Smartphone Use Content

We measured the types of smartphone use content using a revised version of the Mobile Phone Use Pattern Scale [49,50]. The revised scale gauges 20 types of smartphone use content including playing games, chatting online, browsing the news, making payments, learning through online courses, and so on (see Table A1 in the Appendix A for more information on each type). A total of 15 out of 16 items in the original scale were selected and five widely used types of smartphone content (e.g., “Watch short-form videos”, “Use a smartphone to learn online courses”) were added according to the survey results of internet and APPs use in China National Research Report on Internet Use of Minors in 2020 [51] and the White paper on Chinese iGeneration’s psychology and behavior of smartphone use [52]. The scale uses a 4-point Likert scale ranging from 1 (never) to 4 (frequently). The scale had good reliability (α = 0.916). In this study, each item represents an independent construct of different smartphone use content. Furthermore, the psychological constructs using one-item measurement of the Likert scale are comparable to constructs with multiple-items measurement [53]. Thus, the 20 items measure 20 types of different smartphone use content.

#### 2.2.2. Social Relationships

We regarded parent–child, peer, and teacher–student relationships as social relationships for adolescents. We measured the parent–child relationship via the Parent–Child Relationship Scale [54], which was revised from the Social Relations Network Questionnaire [55]. The revised Parent–Child Relationship Scale includes 11 items (e.g., “Are you satisfied with the relationship with your parents?”). Items are rated on a 4-point Likert scale (1 = not at all, 4 = very), with higher total scores indicating a more harmonious parent–child relationship. The scale had adequate reliability (α = 0.897) and validity (CFI = 0.975, TLI = 0.962, RMSEA = 0.049).

We assessed peer relationships with a modified version of the Peer Relationship Scale [56]. The 4-point Likert scale (1 = not at all, 4 = totally agree) includes 10 items (e.g., “I am satisfied with my relationship with my classmates”), with higher total scores representing better peer relationships. The reliability (α = 0.870) and validity (CFI = 0.974, TLI = 0.954, RMSEA = 0.049) were robust in our study.

We evaluated the teacher–student relationship through the Teacher–Student Relationship Scale. Using the PISA 2015 Assessment and Analytical Framework [57], the scale includes 5 items (e.g., “I get along well with my teacher”) scored on a 4-point Likert scale. A higher total score implies a better teacher–student relationship. The scale’s reliability (α = 0.948) and validity (CFI = 0.981, TLI = 0.961, RMSEA = 0.068) were acceptable.

#### 2.2.3. Control Variables

As previously indicated, the control variates in our study included gender, grade, socioeconomic status (SES), and daily smartphone use time. However, as these datasets were collected as part of the Regional Basic Educational Evaluation Program, which focused on achievement differences of students in different grades rather than of different ages, it did not include information on adolescents’ age. Thus, we used grade level to represent age as a covariate. Furthermore, subjective SES was measured with 1 item—“How would you rate your family’s socioeconomic status in this city?” —with scores ranging from very bad (1) to very good (5). Objective SES was measured by the mean of the standardized scores of annual family income and the educational levels of parents [58]. The smartphone use time was averaged by weekday and weekend use time, which ranged from no use (score 0) to 7 h or more (score 7).

### 2.3. Analytic Procedure

First, we pre-processed the data using SPSS 20.0 software, including processing missing values and standardizing each variable for later analysis. Second, we employed SCA to explore the relationship between the types of smartphone content and adolescents’ social relationships through the R package *specr* [59]. Implementation of SCA contains three steps [37]: (1) defining a series of reasonable specifications; (2) estimating all specifications and reporting descriptive results; and (3) making statistical inferences. In the first step, all non-redundant combinations of diverse variables (the dependent, independent, and control variables) should be listed. In the second step, the researcher needs to calculate the predicted regression coefficient for different specifications (combinations), and describe the distributions of the regression coefficients. In the final step, the researcher should implement a statistical inference to determine the extent to which the results are inconsistent with the null hypothesis of no effect. To generate a null hypothesis, one needs to first generate null data by forcing the null on the existing data [37]. In this study, the regression coefficient of the variable of interest, multiplied by the independent variable (types of smartphone use consent), was subtracted from the dependent variable (social relationships). This created a new dependent variable and then the null data was created where we know the null hypothesis is true (namely, there is no relation between types of smartphone use content and mental health in the null data). Then participants were drawn at random, with replacement, from the null data, creating a new SCA model under the null hypothesis. After 500 cycles of the drawing process, we formed the SCA models with the effect of smartphone use content on mental health under the null hypothesis. We then examined whether the statistical indicators from the original SCA were significantly different from the indicators in the bootstrapped SCAs [37]. In the statistics inference section, as previously suggested [37,60], we used the median effect estimate (the median *β*) and the number of significant results in the predominant direction (the NSRPD) to determine whether such predictions of specifications were significant. Third, we calculated effect sizes (partial *r*^2^) for each median regression coefficient in SPSS 20.0. Following the effect size rule when examining the effect of technology use, an effect size (partial *r*^2^) greater than 1% is sufficient to attract the attention of the public and policy-makers [60,61].

## 3. Results

### 3.1. The Identification and Implementation of Specifications

The sum of the identification specifications was 420 (identified specifications= types of content (20 choices) × social relationships (three choices) × control variables (seven choices). As shown in Figure 2, each dot in the top panel (Figure 2A) represents an estimate from a sort of specification; the vertical below each dot (Figure 2B) indicates the estimate of analytic decisions [37]. The regression coefficients ranged from −0.20 to 0.20. We found 159 significantly negative associations, 24 non-significant associations, and 237 significantly positive associations between the types of content and social relationships.

Additionally, we analyzed the result of each social relationship. A total of 92 significantly positive associations, 44 significantly negative associations, and three non-significant associations between types of content and the teacher–student relationship were found. As for the parent–child relationship, 61 associations were positively significant, which is the fewest among the three social relationships, and 75 significantly negative associations and four non-significant associations were found. For peer relationships, 84 associations showed a significantly positive relationship with types of smartphone use content. While 39 significantly negative associations were found, which was the fewest among three social relationships. The remaining 17 associations were non-significant.

### 3.2. Statistical Inferences

As for using smartphones for playing games (see Table 2) (median *β* = −0.12, *p* < 0.001, partial *r*^2^ = 0.01), watching clips (median *β* = −0.08, *p* < 0.001, partial *r*^2^ = 0.006), reading online novels (median *β* = −0.09, *p* < 0.001, partial *r*^2^ = 0.007), and consuming online (median *β* = −0.07, *p* < 0.001, partial *r*^2^ = 0.004) were negatively correlated with interpersonal relationships. These types of smartphone use content may have a negative effect on social relationships and all of their NSRPD were significant (see Table 2). Using smartphones to make calls (median *β* = 0.09, *p* < 0.001, partial *r*^2^ = 0.007), listen to music (median *β* = 0.02, *p* < 0.001, partial *r*^2^ = 0.00), browse the news (median *β* = 0.08, *p* < 0.001, partial *r*^2^ = 0.007), take online courses (median *β* = 0.13, *p* < 0.001, partial *r*^2^ = 0.016), use search engines (median *β* = 0.11, *p* < 0.001, partial *r*^2^ = 0.012), use dictionaries (median *β* = 0.11, *p* < 0.001, partial *r*^2^ = 0.013), use utilities (median *β* = 0.06, *p* < 0.001, partial *r*^2^ = 0.004), use cameras (median *β* = 0.08, *p* < 0.001, partial *r*^2^ = 0.007), use fitness apps (median *β* = 0.09, *p* < 0.001, partial *r*^2^ = 0.008), and obtain life information (median *β* = 0.12, *p* < 0.001, partial *r*^2^ = 0.015) were positively and significantly related to adolescents’ interpersonal relationships. These types of smartphone use content may have a positive effect on social relationships and all of their NSRPD were significant (see Table 2). In addition, using smartphones to browse social media (median *β* = −0.02, *p* > 0.05, partial *r*^2^ = 0.00), post or share information (median *β* = −0.0003, *p* > 0.05, partial *r*^2^ = 0.00), and chat online (median *β* = −0.01, *p* > 0.05, partial *r*^2^ = 0.00) showed no significant correlation with adolescents’ interpersonal relationships.

Furthermore, watching TV and making payments only showed significant correlations with interpersonal relationships in median *β* but not in NSRPD. Of all the above coefficients, only playing games (partial *r*^2^ = 0.01), taking online courses (partial *r*^2^ = 0.016), using search engines (partial *r*^2^ = 0.012), using dictionaries (partial *r*^2^ = 0.013), and obtaining life information (partial *r*^2^ = 0.015) had an effect size equal to or greater than 1%.

## 4. Discussion

In concentrating on the types of smartphone use content, we have challenged the previous replacement theory and the enhancement theory of internet use. Results revealed that using smartphones for interpersonal interactions (e.g., browsing social media) was unrelated to adolescents’ social relationships, and using smartphones for recreation (e.g., playing games, watching clips, or reading online novels) and online consumption (e.g., making payments) were negatively associated with adolescents’ social relationships. However, using smartphones for studying (e.g., taking online courses, using dictionaries) and improving one’s quality of life (e.g., obtaining life information) were positively associated with adolescents’ social relationships.

The frequency of interpersonal-related content used by adolescents during smartphone use was uncorrelated with adolescents’ social relationships, which contradicts Hypothesis 1. This result was also somewhat inconsistent with previous findings, which showed that communication of technology use has a negative or positive effect [19,27,62]. This finding challenges the enhancement theory, which suggests that smartphone use may provide individuals with more opportunities to interact with others and subsequently improve their social bonds [13,14]. One possible explanation is that although focusing on interpersonal communications (such as social media apps or social networking sites [SNS]) can increase the number of opportunities to interact with others, there may also be a potential risk of developing social media dependency or SNS addiction [63,64,65]. In this way, adolescents may develop a tendency to engage in online social interactions with broader social networks or even engage in other online activities unrelated to social interactions [65], so that their offline interpersonal relationships may be compromised. This could also explain why we found the use of smartphones to make calls was positively associated with social relationships, as adolescents might only focus on communication and not access other potentially distracting or addictive content. This is consistent with previous research that communication improves adolescents’ relationships [14,15]. However, even if there was a negative correlation between social media content (e.g., chatting online) regarding smartphone use and adolescents’ interpersonal relationships, the effect size of this relationship is too low to consider further [60,61].

Entertainment content (watching TV, watching clips, playing games, reading online novels) and online transaction content (online consumption, making payments) during smartphone use were negatively related to adolescents’ social relationships. This was consistent with Hypothesis 2 and the previous results, which found that time spent on entertainment use was associated with negative interactions with parents [27]. It is widely believed that using smartphones for entertainment content is related to one’s proneness to problematic smartphone use [50], which may reduce adolescents’ social competence and decrease their interactions with others. However, the most important fact is that the effect sizes described for the correlation between entertainment activities on smartphones and adolescents’ social relationships—except for playing games on smartphones—were relatively low (lower than 1%). The correlation between smartphone gaming use and adolescents’ social relationships presented an effect size of 1%. This might be related to the addictive tendency of online games. One key diagnostic criterion for the Internet gaming disorder, proposed in the DSM-5 and ICD-11, is that it endangers, or causes the loss of, a significant interpersonal relationship [66,67]. Hence, the use of smartphones for gaming should be brought to the attention of both the public and policymakers.

Most importantly, our largest finding was that taking online courses and using helpful tools (using search engine dictionaries, utilities, cameras, fitness apps, and obtaining life information) were positively related to adolescents’ interpersonal relationships. These results failed to support Hypothesis 2, but plausibly supported the enhancement theory of internet use. First, spending less time on social media use or mobile gaming could help adolescents avoid developing social media dependence or smartphone dependency, which would threaten their interpersonal functions [11]. Second, adolescents have more opportunities to interact with their teachers, classmates, and peers when focusing on learning through online courses, which may facilitate good interpersonal relationships and improve their interpersonal skills. Third, learning via online courses, using dictionaries, and using search engines through smartphones may show that adolescents enjoy studying and are willing to engage in academia. According to the 2019 Chinese Internet Users’ Search Engine Usage Research Report [68], 97.1% of internet users used search engines via their smartphones, and the most common usage scenario for search engines was working and studying, at 76.5%. Online learning-related behaviors are consistent with the expectations of parents, teachers, and adolescents themselves, especially in the context of Asian cultures, which place extreme emphasis on academic performance and expect academic achievement from children and adolescents alike [69,70,71]. Once adolescents meet the expectations demanded by their cultures, they are more likely to establish good social bonds with others. Nonetheless, using utilities, cameras, and fitness apps showed relatively low effect sizes (lower than 1%). In addition, the effect sizes between homework completion and social relationships in both samples were too low to be meaningful.

Furthermore, these findings could be interpreted in terms of the social shaping of technology [72], which suggests that the consequences of technology arise from a mixture of “affordances” (social capabilities enabled by technological qualities) and the unexpected and emergent ways in which people use these affordances [41]. That is, the role of technology depends not only on its technical qualities and corresponding social functions, but also on people’s subjective motives and selectivity in using it. From this angle, the impact of smartphone use on adolescents’ social ties should not be generalized to the effect of smartphones’ overall technological qualities, but should be analyzed given what types of smartphone content adolescents choose to use. Our results were in line with this perspective. Additionally, our findings suggested that the theory of the social shaping of technology can be more flexible and practical when examining the effects of smartphone use on adolescents’ social relationships, and can be a good complement to the ongoing debate between displacement and enhancement theories of technology.

Our study has several implications and limitations. We used SCA to empirically test the scientific validity of enhancement and replacement theories from the perspective of different types of content during smartphone use. This may help to supplement the previous debate on whether the use of media technologies is positively related to adolescents’ interpersonal relationships according to the enhancement theory, or negatively correlated with adolescents’ interpersonal relationships based on the replacement theory. Given the two large samples, we argue that the impact of technology use on adolescents’ interpersonal relationships depends on what kinds of content they choose when using their smartphones. For example, focusing on social media, entertainment, and transaction content during smartphone use may follow the rules of the replacement theory, while focusing on online courses or studying content might follow the rules of the enhancement theory. These findings showed that future research should pay more attention to the impact of different types of smartphone use content on individuals’ social relationships. In addition, among the 20 types of smartphone use content, we identified several categories that deserve the attention of the public and policy-makers as they account for at least 1% of the covariance in the magnitude of their effects on adolescents’ interpersonal relationships. Future policies should pay considerable attention to the negative impact of playing games via smartphones on adolescents’ social relationships; they should also recognize the advantages of technology when adolescents focus on activities related to their studies (e.g., taking online courses or using search engines for learning purposes).

The limitations of our study are as follows. First, the analysis was based on cross-sectional datasets, and the results made it difficult to identify a causal direction. There may be a bidirectional association between types of smartphone use content and adolescents’ social relationships. Future researchers could focus on longitudinal or cohort designs to explore the association between smartphone content types and adolescents’ social ties. Second, except for peers, parents, and teachers, we did not include other agents playing roles in adolescents’ socialization process, such as siblings. Although the analyzed datasets did not include other agents, parents, peers, and teachers are three extremely important entities in adolescents’ socialization process [5]. Thus, to a large extent, parent–child, peer, and teacher–student relationships can represent adolescents’ social relationships. Third, in terms of the covariates, we used a grade-level variable instead of age as the study is a secondary analysis of existing datasets that did not contain an age variable due to the unique demands of the RAEQ. Future studies are supposed to control actual age as a covariable. Fourth, the problematic or compulsive use of smartphones may have an effect on the types of smartphone use content and social relationships [38,39,73]. Researchers can include the problematic or compulsive use of smartphones in the future to explore the relationship between types of smartphone use content and social relationships. Finally, our results indicated that different types of social relationships showed different numbers of significant or non-significant results, which indicate that the effect of different types of smartphone use content may exhibit differently according to the types of social relationships. Future studies can further explore this.

## 5. Conclusions

Different types of smartphone use content have different relationships with adolescents’ social relationships. Taking online courses, using search engines, using a dictionary, and obtaining life information via smartphones positively explain at least 1% of the variance of their social relationships, while playing games via smartphones negatively explains 1% of the variance of their social relationships. Regardless of whether smartphone use outcomes follow the rules of the enhancement or displacement theories, the decisive aspect may be the types of content youth choose when using smartphones. Among these types of smartphone use content, the negative effect of playing games, and the positive effect of learning and other activities through smartphones, should be brought to the attention of both the public and policy-makers. 

## Figures and Tables

**Figure 1 ijerph-19-08940-f001:**
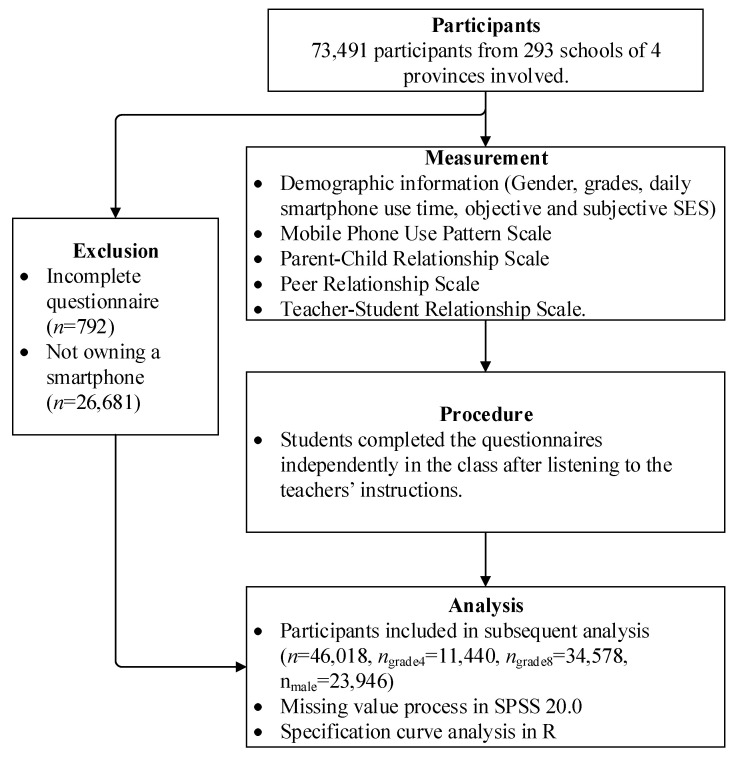
Overall research methodological procedure.

**Figure 2 ijerph-19-08940-f002:**
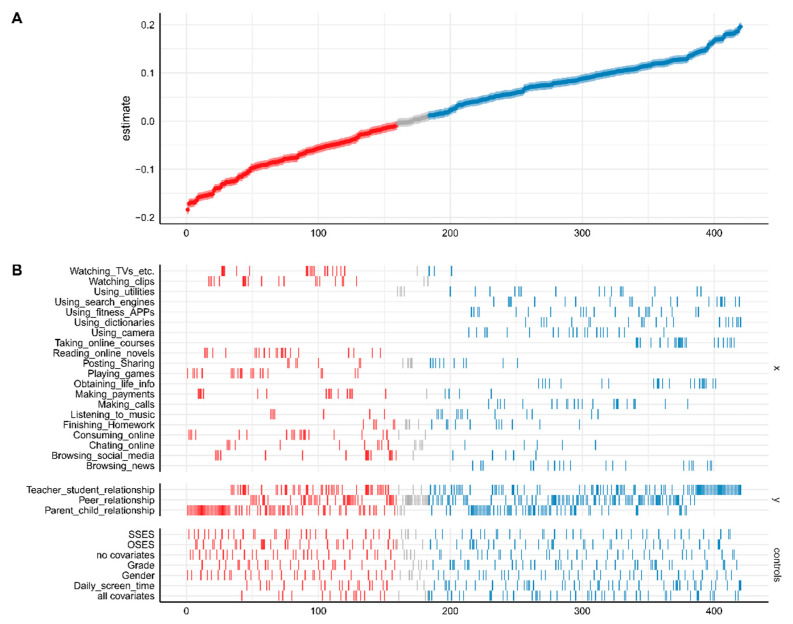
Description of SCA. The estimate denotes the standardized regression coefficients calculated via linear regression; the *x*-axis indicates the combined number of specifications, and the *y*-axis of (**B**) refers to all variables used in the analysis. Red dots denote negative and significant effects (based on the chosen significance level; by default, α = 0.05). Blue dots signal positive and significant effects. Gray dots indicate insignificant effects. The coefficients in (**A**) depict a curve, and the dashed area around the curve denotes the 95% CIs for the standardized regression coefficients. Daily screen time means smartphone use time each day; SSES is subjective SES, and OSES is objective SES.

**Table 1 ijerph-19-08940-t001:** Participants characteristic of grade 4 and grade 8 students.

Variables	Groups	Grade 4	Grade 8
Gender	Boys	52.3%	51.9%
	Girls	47.7%	48.1%
Residence	City	29.7%	26.3%
	Rural region	70.3%	73.7%
Single parent	Yes	6.6%	4.6%
	Not	93.4%	95.4%
Only child	Yes	26.5%	19.3%
	Not	73.5%	80.7%
Left-behind child	Yes	16.4%	9.4%
	Not	83.6%	90.6%
Annual revenue	<60,000 ¥	55.0%	59.9%
	60,000–100,000 ¥	27.2%	22.2%
	>100,000 ¥	17.9%	17.9%

Note: ¥ = RMB.

**Table 2 ijerph-19-08940-t002:** The results of SCA for the different types of smartphone use content and social relationships.

Independent Variables	Median β	Partial *r^2^*	NSRPD
Making calls	0.09 ***	0.007	21/21 ***
Browsing social media	−0.02	0.00	15/21
Posting/sharing	−0.0003	0.00	7/21
Chatting online	−0.01	0.00	11/21
Watching TV, etc.	−0.06 ***	0.003	17/21
Watching clips	−0.08 ***	0.006	19/21 ***
Playing games	−0.12 ***	**0.01**	21/21 ***
Listening to music	0.02 ***	0.00	14/21 ***
Browsing news	0.08 ***	0.007	21/21 ***
Reading online novels	−0.09 ***	0.007	21/21 ***
Taking online courses	0.13 ***	**0.016**	21/21 ***
Finishing homework	0.01 ***	0.001	11/21 ***
Using search engines	0.11 ***	**0.012**	21/21 ***
Using dictionaries	0.11 ***	**0.013**	21/21 ***
Using utilities	0.06 ***	0.004	17/21 ***
Using cameras	0.08 ***	0.007	21/21 ***
Using fitness apps	0.09 ***	0.008	21/21 ***
Obtaining life information	0.12 ***	**0.015**	21/21 ***
Making payments	−0.05 ***	0.002	16/21
Consuming online	−0.07 ***	0.004	19/21 *

Note. * *p* < 0.05, *** *p* < 0.001, NSRPD = Number of significant results in predominant direction; Median *β* = Median point estimate. Partial *r^2^* equal to or greater than 0.01 are bold.

## Data Availability

The datasets generated during and/or analyzed during the current study are available from the corresponding author on reasonable request and with permission of Collaborative Innovation Centre of Assessment toward Basic Education Quality, Beijing Normal University, Beijing, China.

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
