# Peer review of "Beyond Screen Time: Exploring the Associations between Types of Smartphone Use Content and Adolescents’ Social Relationships"

_ijerph, 2022, doi:10.3390/ijerph19158940_

Round 1

Reviewer 1 Report

Thank you for the opportunity to review this excellent and timely paper. The paper is well written and straightforward to read, and the narrative is sound. My only suggestion is to consider the redundant values reported in section 3.2 and Table 2 - it seems as though these two points could be integrated into a single larger table to make it a little clearer and save some space. This is a very minor point, though, and I was very impressed by the paper overall!

Author Response

Dear Reviewer,

We deeply appreciate your valuable comments on the manuscript entitled “Beyond Screen Time: Exploring the Associations between Types of Smartphone Use Content and Adolescents’ Social Relationships” (MS number ijerph-1800254).

We deeply appreciate your comments and suggestions. We have modified our description of the SCA result. The NSRPD indicators were all deleted from the text. But for readability, we kept the β, p value, and partial r2 indicators.

Again, we thank you for providing constructive feedback!

Reviewer 2 Report

The topic is interesting and important. However, there are several key areas
that need more work prior to publication. I have summarized the required
changes in the hope that the feedback will be useful to you as you update
the paper.

The authors should ask the help of native English speaking proofreader,
because there are some many typo and linguistic mistakes that should be
fixed.
The introduction is poorly written and it does not properly refer to
previously published studies. The authors need to carefully review the
published literature, identify the gaps in the literature, and propose their
approach to fill the gap.

The literature review is not enough. It is important to add some recent work
(2018-2022) to the literature review. At least 10 new references should be
added to the article.

A flowchart should be added to the article to show the research
methodology.

Much more explanations and interpretations should be added for the
result, which are not enough.
It is suggested to compare the results of the present study with previous studies and analyze their results completely.

Author Response

Dear reviewer, 

    We thank your valuable comments and suggestions. We submitted our responses to your comments in the attached file. Please see the attachment.

    Best wishes !

Reviewer 3 Report

This is an interesting study, with an approach that attempts to advance the partial understanding of the problems associated with smarthphones. It proposes the analysis of content typologies, as opposed to the mere frequency and time of use. However, I would like to point out to the authors several issues that should be considered in their manuscript.

Abstract

The instrument used to evaluate the study variables should be indicated.

Introduction

Some of the bibliographical references used are very old. This can be justified in specific cases in which the authors are referents, but this is not fulfilled in most cases, nor should it be what predominates. Bibliographic references should be updated with more than 5-7 years. From a study focused on topical issues, it is expected that it will be prepared on the basis of the most current references possible (2-3 years).

Materials and Methods

The procedure for accessing the sample and its characteristics should be described in greater detail. It should also be indicated if it is a cross-sectional study and if the sample is of convenience, etc.

In relation to the instrument used to evaluate the content of use of smarthphones, the authors refer to two previous studies of themselves. However, these references do not justify the desirability of such an instrument per se. Please indicate how it is constructed and on the basis of which instruments or previous studies it was made. Such self-criticism may also be considered inappropriate if its use is not duly justified.

Not having the age of the participants, and using the grade level instead to represent the age as covariable, is a limitation that should be indicated. There may have been students who have repeated courses, whose age is higher than the one assumed by the level of degree in which they are. Likewise, the rate of repeaters is not reported, because the extent of this bias is unknown.

It should be clearly stated how the SES was assessed subjectively and objectively.

The authors mention previous literature related to how SES can influence the use of smarthphones. Also about the relationship of screen time with well-being and satisfaction with life. But these relationships are approached superficially. It is suggested to expand the bibliography in this regard, and incorporate it in the Introduction section. In such a way that, in the subsection of Variables, it is indicated in a concise way how these variables were evaluated.

It is necessary to have evaluated a possible problematic or compulsive use of smathphones, which may be influencing the content accessed, and the impact on social relations. If the authors evaluated something in this sense, it may be interesting to include it as covariable.

In the Introduction section, the authors have also explicitly pointed out the need to investigate the association between different types of content in the use of smarthphones, and the main social relations of adolescents: parents-children; peer-to-peer; teacher-learner. However, in the analysis of content types they have been limited to only two typologies (related to interpersonal interactions; and related to non-internative interactions). This fact, in itself, is already limiting for the contribution of novel findings. Is it possible to subdivide these two typologies into more specific typologies that provide more novel information? In principle, the population sample of the study appears to be large enough to present statistical power.

On the other hand, the three main social relations referred to are reflected in the identification and implementation of specifications. However, no analysis is presented to establish how the use of each type of use affects each of these three social relations.

Also, considering what the specialized literature indicates regarding the differences in the use and content of boys and girls, it is considered appropriate to make these analyses according to the gender of the participants. In fact, the authors themselves point this out in the Introduction.

Results

Figure 1 incorporates two images. It is suggested that these be presented independently.

It is suggested to revise the title of table 2.

Information on how the SES and screen time were evaluated is missing to consider the findings presented. Also the additional analyses noted in the previous paragraph.

Discussion

The discussion is well presented from the results presented. However, some of the questions raised would be satisfied if the above analyses were carried out (on problematic or compulsive use of smathphones; gender-disaggregated analysis; how it relates to each of the three main social relationships in adolescents). For this reason, it is considered insufficient to offer truly novel contributions for researchers, professionals and the general public.

Author Response

(The authors gave the same response as above.)

Round 2

Reviewer 2 Report

The section of conclusion is still short. Some more concluding remarks should be added.

Author Response

Dear reviewer,

Thank you for your further suggestion. We extended the conclusion part according to your suggestion. The revised part was highlighted in the manuscript.

The conclusion of the manuscript was revised as “Different types of smartphone use content have different relationships with adolescents’ social relationships. Taking online courses, using search engines, using a dictionary, and obtaining life information via smartphones positively explain at least 1% of the variance of their social relationships, while playing games via smartphones negatively explains 1% of the variance of their social relationships. Regardless of whether smartphone use outcomes follow the rules of the enhancement or displacement theories, the decisive aspect may be the types of content youth choose when using smartphones. Among these types of smartphone use content, the negative effect of playing games, and the positive effect of learning and other activities through smartphones, should be brought to the attention of both the public and policy-makers.”

Besides, the English language and style are checked again, some grammar or spelling errors were corrected and highlighted in the manuscript.

Best wishes!

Reviewer 3 Report

The authors have largely heeded the suggestions and comments made. The manuscript has improved considerably.

Author Response

Dear reviewer,

    We thank your valuable suggestions in the first round, which help improve our manuscript a lot. Besides, we checked the English language and style of our manuscript again, some grammar errors were corrected and highlighted in the manuscript. 

Best wishes!